# Peroral Cholangioscopy-Guided Targeted Biopsy versus Conventional Endoscopic Transpapillary Forceps Biopsy for Biliary Stricture with Suspected Bile Duct Cancer

**DOI:** 10.3390/jcm11020289

**Published:** 2022-01-06

**Authors:** Katsunori Sekine, Ichiro Yasuda, Shinpei Doi, Noriyuki Kuniyoshi, Takayuki Tsujikawa, Yuichi Takano, Masatoshi Mabuchi, Kosuke Takahashi, Masashi Kawamoto, Mikiko Takahashi, Tatsuya Aso, Tatsuhiko Miyazaki, Takuji Iwashita

**Affiliations:** 1Department of Gastroenterology, Teikyo University Mizonokuchi Hospital, Kawasaki 213-8507, Japan; ka2nori924@gmail.com (K.S.); sinpesan@gmail.com (S.D.); s06030kn@yahoo.co.jp (N.K.); tsujikawa19820506@gmail.com (T.T.); yuichitakano1028@yahoo.co.jp (Y.T.); masatoshi.mabuchi@gmail.com (M.M.); 2Third Department of Internal Medicine, University of Toyama, Toyama 930-0194, Japan; takapochi0809@gmail.com; 3Department of Diagnostic Pathology, Teikyo University Mizonokuchi Hospital, Kawasaki 213-8507, Japan; kugenuma1320@gmail.com (M.K.); mitakahashi@med.teikyo-u.ac.jp (M.T.); taso@med.teikyo-u.ac.jp (T.A.); 4Department of Pathology, Gifu University Hospital, Gifu 501-1194, Japan; tats_m@gifu-u.ac.jp; 5First Department of Internal Medicine, Gifu University Hospital, Gifu 501-1194, Japan; takuji@w7.dion.ne.jp

**Keywords:** POCS, SpyGlass, transpapillary biopsy, biopsy sample

## Abstract

Background: The recent improvement of peroral cholangioscopy (POCS) maneuverability has enabled the precise, targeted biopsy of bile duct lesions under direct cholangioscopic vision. However, as only small-cup biopsy forceps can pass through the scope channel, the resulting small sample size may limit the pathological diagnosis of biopsy specimens. This study compared the diagnostic abilities of POCS-guided biopsy and conventional fluoroscopy-guided biopsy for bile duct cancer. Method: This multicenter, retrospective cohort study included patients exhibiting bile duct stricture with suspected cholangiocarcinoma in whom POCS-guided and fluoroscopy-guided biopsies were performed in the same session. The primary endpoint was the diagnostic sensitivity for malignancy. The size and quality of the biopsy specimens were also compared. Result: A total of 59 patients were enrolled. The sensitivity of POCS-guided biopsy was similar to that of fluoroscopy-guided biopsy (54.0% and 64.0%, respectively). However, when the modalities were combined, the sensitivity increased to 80.0%. The mean specimen size from POCS-guided biopsy was significantly smaller than that from fluoroscopy-guided biopsy. The specimen quality using fluoroscopy-guided biopsy was also better than that using POCS-guided biopsy. Conclusions: The diagnostic sensitivity of POCS-guided biopsy is still insufficient, mainly because of the limited specimen quantity and quality. Therefore, conventional fluoroscopy-guided biopsy would be helpful to improve diagnostic sensitivity.

## 1. Introduction

The incidence of bile duct cancer is increasing globally. The current non-invasive diagnostic approaches are not accurate enough, and pathological confirmation is necessary. However, the highly desmoplastic nature of bile duct cancer limits the accuracy of non-surgical pathological approaches. In addition, the efficacy of the available therapies is compromised by their pathological nature and high genomic heterogeneity. Currently, molecular profiling of bile duct tumors is becoming popular, and it may alter the treatment strategy of these tumors in the near future. Therefore, there is an urgent need to collect sufficient tissue samples for the accurate diagnosis and molecular profiling of these tumors [1,2]. Pathological sampling for the diagnosis of bile duct cancer is generally performed using conventional endoscopic transpapillary forceps biopsy, brushing cytology, or bile juice cytology using endoscopic retrograde cholangiopancreatography (ERCP) under fluoroscopic guidance. However, the sensitivity of these methods is relatively low. Recently, targeted biopsy under the direct vision of peroral cholangioscopy (POCS) was introduced. Many studies have reported that POCS-guided targeted biopsy has a higher sensitivity than a fluoroscopy-guided biopsy and brushing cytology. However, the specimens obtained by POCS-guided biopsy are usually small, and pathological diagnosis is often difficult. Further, in our clinical experience, fluoroscopy-guided biopsy sometimes provides better results. To elucidate the real utility of POCS-guided biopsy, we conducted a retrospective study to compare the diagnostic ability of POCS-guided biopsy with that of fluoroscopy-guided biopsy in patients with suspected bile duct cancer. In addition, the size and quality of the biopsy samples were compared between the two biopsy methods.

## 2. Materials and Methods

### 2.1. Study Design

This retrospective, multicenter cohort study was conducted at the Teikyo University Mizonokuchi Hospital, the Gifu University Hospital, and the University of Toyama Hospital. Written informed consent to the procedure was obtained from all patients before the procedure was initiated. The study protocol was approved by the institutional review board of each participating institution (TUIC-COI 18-0059, R2020058) and was registered in the University Hospital Medical Information Network (UMIN) clinical trial registry (UMIN ID: UMIN00040844).

### 2.2. Patients

Between November 2013 and February 2020, 59 patients underwent both POCS-guided and fluoroscopy-guided biopsy of extrahepatic biliary stricture for suspected bile duct cancer in the same session. In all patients, a POCS-guided targeted biopsy was initially performed, followed by a fluoroscopy-guided biopsy, because bleeding after the fluoroscopy-guided biopsy made the cholangioscopic view unclear. We reviewed the clinical and laboratory data from the ERCP database and patients’ medical records.

### 2.3. Diagnostic Algorithms of Suspected Bile Duct Cancer in Our Institutions

In cases of suspected biliary stricture based on blood examinations and ultrasonography, multidetector-row computed tomography (MD-CT) was performed in our institutions. If the image findings indicated bile duct cancer, ERCP-related procedures, including pathological sampling and subsequent biliary drainage, were attempted. If the MD-CT results did not show any findings suspicious of unresectable tumors, and the distal end of the biliary stricture did not reach the papilla, POCS with targeted biopsy was attempted. A fluoroscopy-guided biopsy was performed following the POCS-guided biopsy, as described above.

### 2.4. Endoscopic Procedures

All endoscopic procedures were performed by one of four experienced endoscopists who have each performed more than 1000 ERCP procedures (K.S., I.Y., S.D. and T.I.). The procedures were performed under suitable sedation using midazolam, pentazocine, and dexmedetomidine. During the procedure, pulse rate, oxygen saturation, electrocardiography, and blood pressure were continuously monitored.

First, a conventional duodenoscope (TJF-260V; Olympus, Tokyo, Japan) was advanced to the second part of the duodenum. After successful biliary cannulation and placement of a guidewire into the bile duct, a cholangiogram was obtained. Endoscopic sphincterotomy was performed prior to the POCS. A single-operator POCS (SpyGlass Legacy or DS, Boston Scientific Inc., Natick, MA, USA) was then advanced over the guidewire through the 4.2 mm working channel of the duodenoscope. The SpyGlass system is composed of two modularized components: a sterile, single-use cholangioscope with a 1.2 mm diameter working channel and a combined image processor and light source. After careful observation of the lesion using the cholangioscope, POCS-guided tissue sampling was performed on the targeted biliary lesion using dedicated biopsy forceps (SpyBite Biopsy Forceps, Boston Scientific Inc.) under direct cholangioscopic vision (Figure 1).

Following the POCS-guided targeted biopsy, the cholangioscope was withdrawn from the duodenoscope, and the fluoroscopy-guided transpapillary tissue sampling was performed using conventional biopsy forceps (Radial Jaw 4P; Boston Scientific Inc., Natick, MA, USA) through the working channel of the duodenoscope (Figure 2). The cup sizes of the biopsy forceps are shown in Figure 3. Each biopsy from a bile duct stricture suspected of biliary malignancy was performed a minimum of two times. However, the actual number of biopsy samples obtained was determined by the endoscopist based on the macroscopic volume of the biopsy sample and the patient’s condition.

Forceps with a 1.0 mm diameter smooth-shaped cup (SpyBite, Boston Scientific Inc., Natick, MA, USA) (upper) were used for the cholangioscopy-guided biopsy, and forceps with a 1.8 mm diameter jagged-shape cup (Radial Jaw 4P, Boston Scientific Inc.) (lower) were used for the fluoroscopy-guided biopsy.

### 2.5. Final Diagnosis

For patients who underwent surgery, the final diagnosis was made based on the pathological evaluation of the tissue samples obtained during surgery. Patients who did not undergo surgery were followed up for at least 6 months based on the laboratory and imaging tests, regardless of the pathological results (benign or malignant) of the biopsy samples, and the final diagnosis (benign or malignant) was determined based on the clinical course, including laboratory and imaging tests.

### 2.6. Outcome Measurements

The primary outcome measurements were the sensitivities of POCS-guided biopsy and fluoroscopy-guided biopsy for the diagnosis of biliary stricture. Secondary outcomes were the accuracy, specificity, positive predictive value (PPV), negative predictive value (NPV), size of the biopsy sample, and sample quality for each modality.

### 2.7. Evaluation of Biopsy Samples

POCS-guided and fluoroscopy-guided biopsy samples were evaluated by three experienced pathologists (M.K., M.T. and T.A.). Each pathologist made a diagnosis and independently evaluated the quality of the biopsy samples. Each pathologist was blinded to the pathological diagnosis and quality evaluation of the other pathologists until all samples had been evaluated. The pathological quality of the biopsy sample was categorized into the following four grades: “inadequate”, indicating an inadequate sample for histopathological interpretation; “poor”, indicating an insufficient sample for definite histopathological interpretation; “good”, indicating a fair sample for histopathological interpretation; and “excellent”, indicating a good sample with sufficient quantity for definite histopathological interpretation. When multiple specimens were collected from the same patient, the one with the best quality was selected to represent the case. When two pathologists disagreed regarding the quality, the majority judgment was determined to be the final quality; if all the pathologists disagreed, the quality was decided after consultation between the three pathologists. The sizes of the biopsy samples were measured and calculated by a pathologist using a microscope (BX50, DP21; Olympus).

### 2.8. Statistical Analysis

The sensitivity, specificity, PPV, NPV, and accuracy were calculated with a 95% confidence interval (CI), and comparisons were performed using McNemar’s test. Continuous variables were presented as the mean with standard deviation, and comparisons were performed using the Mann–Whitney U test. The ordinal variables for pathological assessment were also tested using the Mann–Whitney U test. A *p* value of less than 0.05 was considered to indicate a significant difference. All statistical analyses were performed using Stata 13 software (Stata Corp., College Station, TX, USA).

## 3. Results

### 3.1. Patient Characteristics

A total of 59 patients (40 men and 19 women; median age, 74 years) underwent ERCP with POCS. Both POCS-guided targeted biopsy and fluoroscopy-guided transpapillary biopsy were performed at the extrahepatic biliary stricture lesion in all patients. The baseline characteristics of the patients are shown in Table 1. The biliary stricture was located in the proximal bile duct in 25 patients (42%) and in the distal bile duct in 34 patients (58%). The median length of the biliary stricture was 16 mm (range, 2–54 mm).

The final diagnoses were bile duct cancer in 48 patients, gallbladder cancer in 2 patients, and benign stricture in 9 patients. The final diagnoses were obtained based on surgical pathology in 33 patients and a combination of biopsy results and the clinical course in 26 patients (range, 5–84 months).

### 3.2. Sensitivity, Specificity, PPV, NPV, and Accuracy of Fluoroscopy- and POCS-Guided Biopsies

The diagnostic results of the POCS-guided and fluoroscopy-guided biopsies are shown in Table 2. The diagnostic sensitivities were 54.0% for POCS-guided biopsy and 64.0% for fluoroscopy-guided biopsy (*p* = 0.416). When a biopsy result from either approach was judged as positive, the sensitivity increased to 80.0%. 

### 3.3. Comparison of the Size of the Biopsy Samples

The number and size of the POCS-guided and fluoroscopy-guided biopsy samples are shown in Table 3. Compared with POCS-guided biopsy, fluoroscopy-guided biopsy provided significantly larger biopsy samples (*p* < 0.001).

### 3.4. Comparison of the Quality of the Biopsy Samples

The pathological assessment of the quality of the biopsy specimens is presented in Table 4. The quality was judged as excellent in 66.1% (39/59) of the fluoroscopy-guided biopsy specimens, whereas excellent samples were obtained in only 35.6% (21/59) of the POCS-guided biopsies. The quality of histopathological specimens was significantly poorer in those obtained using POCS-guided biopsy than in those obtained using fluoroscopy-guided biopsy.

### 3.5. Adverse Events

Cholangitis was observed after the endoscopic procedures in five cases (mild in four and moderate in one). However, this was resolved by fasting combined with a 2–6-day course of antibiotics in all cases. Post-ERCP pancreatitis did not occur in any cases.

## 4. Discussion

Endoscopic transpapillary forceps biopsy under fluoroscopic guidance using ERCP has been widely performed to diagnose indeterminate biliary strictures. The diagnostic sensitivity and specificity for malignancy are reported to range from 43% to 81% and 90% to 100%, respectively [2,3,4,5]. A meta-analysis showed that the pooled sensitivity and specificity were 48.1% (95% CI, 42.8–53.4) and 99.2% (95% CI, 97.6–99.8), respectively [6]. Therefore, the diagnostic specificity is high, but the sensitivity is relatively low. The main reason for this low sensitivity is incorrect biopsy sampling at inadequate sites. To overcome this low sensitivity, targeted biopsy under direct vision of the lesion using POCS was introduced. However, the early model of POCS exhibited several issues, such as fragility and poor maneuverability. However, the new POCS, using the SpyGlass Direct Visualization System, addressed these issues. This improvement enabled precise biopsy sampling at the exact site of the lesion. Many studies have evaluated the diagnostic results of POCS-guided targeted biopsy for indeterminate biliary stricture with reported sensitivities and specificities for malignancies of 64–86% and 89–100%, respectively [4,7,8,9,10,11,12,13,14,15,16,17,18]. Hence, the diagnostic sensitivity of POCS-guided biopsy appears to be superior to that of fluoroscopy-guided biopsy. However, some studies did show relatively low sensitivity [19]. Some previous studies have directly compared the diagnostic results of POCS-guided biopsy with those of conventional endoscopic transpapillary biopsy under fluoroscopic guidance (Table 5) [20,21,22].

One study showed that the diagnostic sensitivity of POCS-guided biopsy was significantly better than that of fluoroscopy-guided biopsy [20]. However, two subsequent studies failed to demonstrate the superiority of POCS-guided biopsy [21,22]. In addition, a meta-analysis reported that the pooled sensitivity of POCS-guided biopsy for malignant biliary strictures was 60.1% (95% CI, 54.9–65.2) [23], which is relatively low. Our early experiences using the POCS-guided targeted biopsy often showed failed diagnostic results and using the fluoroscopy-guided biopsy sometimes provided better results. Therefore, we performed both the fluoroscopy-guided biopsy and the POCS-guided biopsy in routine practice. In the present study, we aimed to elucidate the real benefit of using the POCS-guided biopsy. The diagnostic sensitivity of POCS-guided biopsy was compared with that of fluoroscopy-guided biopsy in patients with suspected bile duct cancer. We found that the diagnostic sensitivity of POCS-guided biopsy was lower than that of fluoroscopy-guided biopsy (54.0% and 64.0%, respectively), although the difference was not statistically significant. The reason for this low sensitivity is likely the limited volume and quality of the biopsy samples. Only mini-cup biopsy forceps can be used during POCS, and these forceps can only obtain small biopsy specimens. Indeed, according to our measurement of the histopathological specimens, the mean size of the POCS-guided biopsy samples was approximately half that of the fluoroscopy-guided biopsy samples. Additionally, the quality of the histopathological samples from the POCS-guided biopsies was significantly poorer than the samples from the fluoroscopy-guided biopsies. The POCS has only a small working channel (1.2 mm diameter), and the dedicated biopsy forceps are much smaller than conventional biopsy forceps (Figure 3). Larger biopsy forceps can sample wider and deeper areas in the bile duct wall (Figure 4). 

The larger biopsy samples obtained from fluoroscopy-guided procedures rather than POCS-guided procedures should make pathological interpretation of biopsy specimens easier and more accurate. Previously, an RCT was conducted to compare the diagnostic sensitivities of fluoroscopy-guided transpapillary biopsy using large-sized forceps (cup size of 2.2 mm) and standard-sized cup forceps (cup size of 1.8 mm) in the diagnosis of extrahepatic biliary stricture. The results showed that the large-sized forceps provided better sensitivity than the standard-sized forceps (70% versus 43%) [24]. Thus, the cup size of biopsy forceps correlates with the quality and quantity of the biopsy specimen. If the same size caliber forceps were available in both procedures, the sensitivity of a POCS-guided biopsy may be better than that of a conventional fluoroscopy-guided biopsy because more precise target biopsies can be collected using a POCS-guided biopsy. Therefore, the development of small-diameter biopsy forceps with a larger cup size or a POCS with a larger biopsy channel is strongly desired. Recently, new biopsy forceps for SpyGlass (SpyBite Max biopsy forceps, Boston Scientific Inc.) became commercially available. These new forceps may improve the quantity and quality of biopsy samples. In addition, large-volume samples may be beneficial for molecular profiling of the tumor, which is becoming important for biliary cancer as well as other cancers.

The present study also revealed that combining a POCS-guided biopsy and a fluoroscopy-guided biopsy improved the diagnostic sensitivity. The sensitivity of the combined approach was 80.0%, whereas the sensitivities of POCS- and fluoroscopy-guided approaches alone were 54.0% and 64.0%, respectively. Therefore, the combination of both biopsy techniques appears to be helpful in obtaining appropriate pathological results.

Regarding the tumor location, the diagnostic sensitivity was not significantly different between distal and proximal bile duct biopsies in either POCS- or fluoroscopy-guided biopsy (POCS-guided, distal, 55.2% versus proximal, 52.4%; fluoroscopy-guided, distal, 62.1% versus proximal, 66.7%).

Finally, this study had several limitations. First, this was a retrospective study. Therefore, there may have been a selection bias in patient enrollment. However, as both POCS-guided and fluoroscopy-guided biopsies were performed on all patients, the baseline characteristics of the patients were identical when comparing the two procedures. Second, the two procedures were performed in the same order in all cases. POCS-guided biopsy was performed first and fluoroscopy-guided biopsy second for all patients. This was because the cholangioscopic view would become unclear due to bleeding and mucosal injury if a fluoroscopy-guided biopsy was performed first. Such unclear vision would have made cholangioscopic observation and subsequent target biopsy quite difficult. However, this order might be disadvantageous for fluoroscopy-guided biopsy. Third, the final diagnoses were made based on surgical pathology in only 33 patients; in the remaining 26 patients, the diagnoses were determined from the biopsy results and the patient’s clinical course. However, the imaging findings and clinical symptoms of 17 of these patients indicated progression with a final diagnosis of malignancy, and all of them died within 5 to 15 months after diagnosis. In contrast, nine patients with a final diagnosis of benign stricture did not show any findings suspicious of malignancy in the imaging, blood tests, or clinical course (follow-up: 17–84 months). To overcome these limitations, randomized controlled studies should be conducted in the future.

## 5. Conclusions

The diagnostic sensitivity of POCS-guided targeted biopsy is still insufficient because of the limited quantity and quality of the biopsy specimens. Therefore, the development of a novel POCS scope with a large working channel to allow the passage of large caliber forceps, or the development of novel biopsy forceps that enable sufficient samples to be obtained is desirable. Until that time, additional conventional fluoroscopy-guided biopsy collection using large-cup biopsy forceps should be considered to improve the diagnostic sensitivity in clinical investigations for biliary tract cancers.

## Figures and Tables

**Figure 1 jcm-11-00289-f001:**
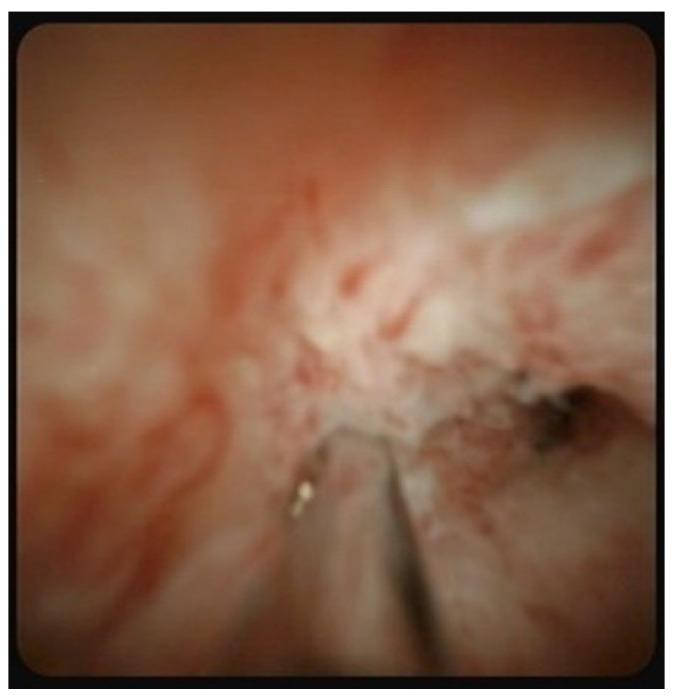
Targeted biopsy under direct vision of peroral cholangioscopy in a case of suspected bile duct cancer.

**Figure 2 jcm-11-00289-f002:**
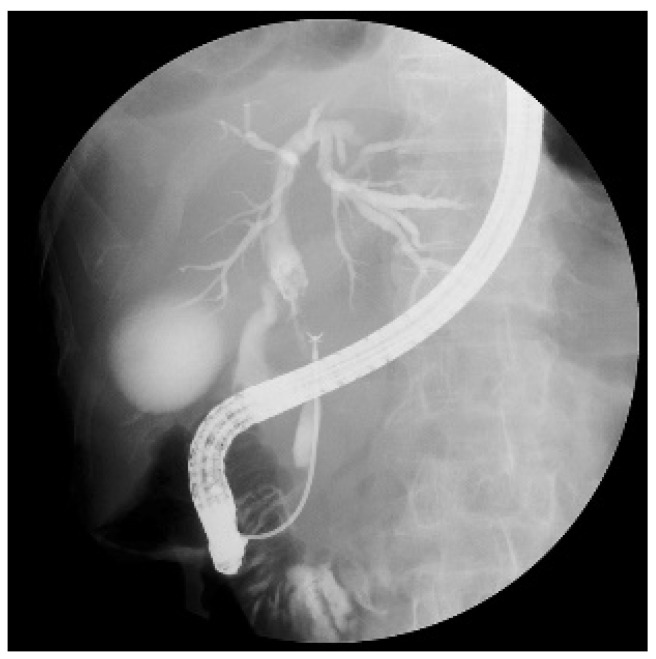
Fluoroscopic image of a fluoroscopy-guided biopsy in a case of suspected bile duct cancer.

**Figure 3 jcm-11-00289-f003:**
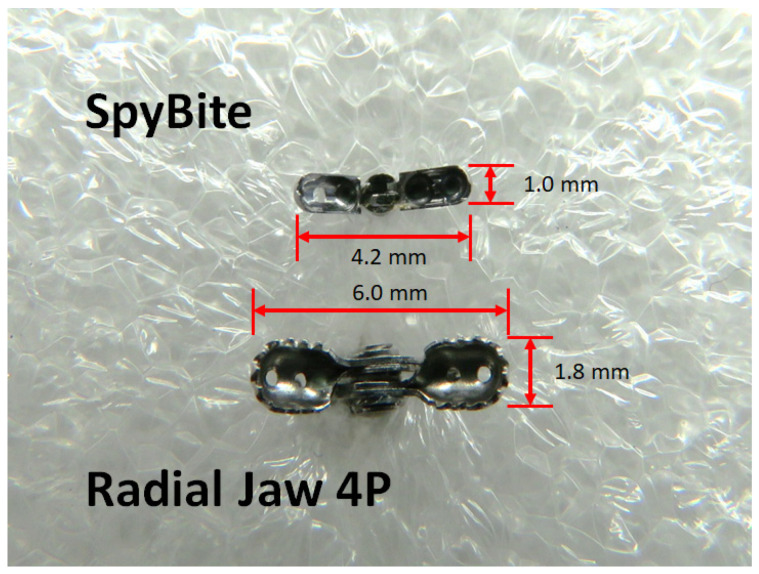
Comparison of biopsy forceps.

**Figure 4 jcm-11-00289-f004:**
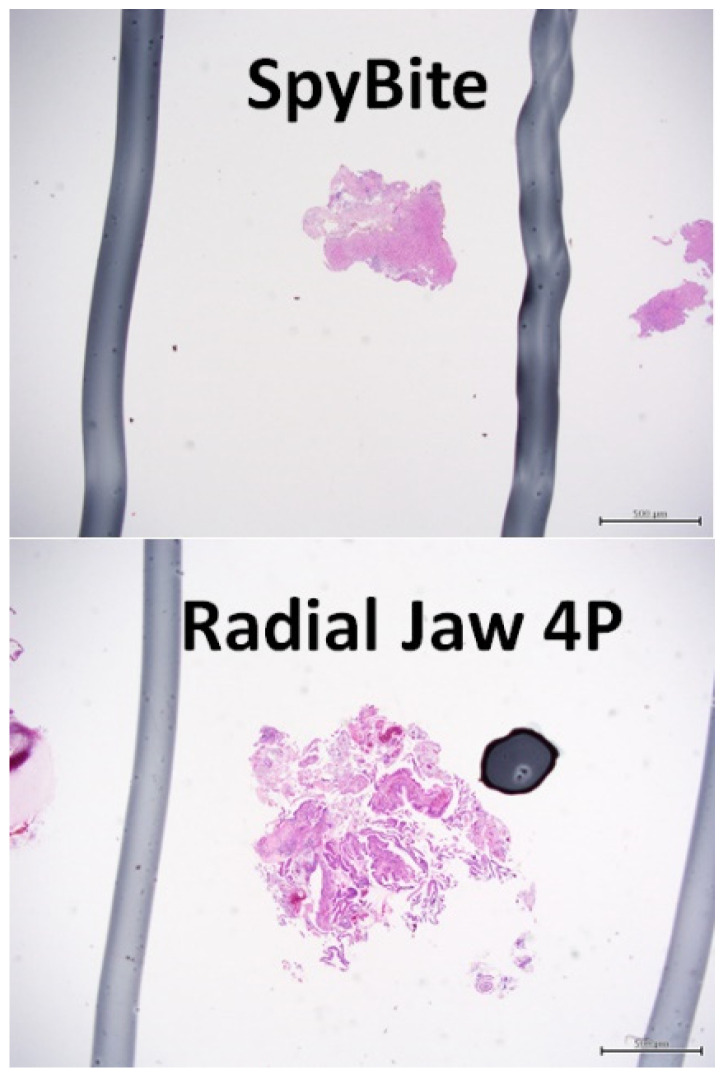
Microscopic images of pathological samples obtained by peroral cholangioscopy-guided and fluoroscopy-guided biopsy. Pathological samples from cholangioscopy-guided biopsies (**upper panel**) and fluoroscopy-guided biopsies (**lower panel**).

**Table 1 jcm-11-00289-t001:** Baseline characteristics of patients (N = 59).

Median Age, Years (Range)	74 (43–89)
Male/female	40/19
Location of the lesion	
Hilar	25
Distal	34
Median length of biliary stricture, mm (range)	16 (2–54)
Final diagnosis	
Bile duct cancer	48
Gallbladder cancer	2
Benign stricture	9

**Table 2 jcm-11-00289-t002:** Comparison of diagnostic results for suspected biliary duct cancer between POCS-guided and fluoroscopy-guided biopsy (n = 59).

	Sensitivity (95% CI)	Specificity (95% CI)	PPV (95% CI)	NPV (95% CI)	Accuracy (95% CI)
POCS-guided biopsy	54.0% (40.4–67.0)	100% (70.1–100)	100% (87.5–100)	28.1% (15.6–45.4)	61.0% (48.3–72.4)
Fluoroscopy-guided biopsy	64.0% (50.1–75.9)	100% (89.3–100)	100% (89.3–100)	33.3% (18.6–52.2)	69.5% (56.9–79.7)
Combined cholangioscopy-guided and fluoroscopy-guided biopsy	80.0% (67.0–88.8)	100% (70.1–100)	100% (91.2–100)	47.4% (27.3–68.3)	83.1% (71.5–90.5)

CI, confidence interval; NPV, negative predictive value; PPV, positive predictive value.

**Table 3 jcm-11-00289-t003:** Number and size of POCS-guided and fluoroscopy-guided biopsy samples.

	POCS-Guided Biopsy	Fluoroscopy-Guided Biopsy	*p*-Value
Number of biopsy samples	2.2 ± 0.7	2.1 ± 0.6	0.163
Size of sample, mm^2^	0.90 ± 1.13	1.77 ± 2.00	<0.001

Mean ± standard deviation.

**Table 4 jcm-11-00289-t004:** Pathological assessment of the quality of biopsy samples.

	Excellent	Good	Poor	Inadequate	*p*-Value
POCS-guided biopsy	21	29	9	0	0.006
Fluoroscopy-guided biopsy	39	14	6	0	

**Table 5 jcm-11-00289-t005:** Comparative studies of POCS-guided targeted biopsy with fluoroscopy-guided biopsy for indeterminate biliary stricture.

Author (Year)	Study Design	Method	N	Sensitivity	*p*
Draganov (2012)	Prospective	Fluoroscopy-guided POCS-guided	26 26 (identical cohort)	29.4% 76.5%	0.0215
Walter (2016)	Retrospective	Fluoroscopy-guided POCS-guided	68 38	45.7% 58.3%	0.674
Onoyama (2020)	Retrospective	Fluoroscopy-guided POCS-guided	31 31 (propensity score-matched cohort)	82.4% 83.3%	1.000
Present study	Retrospective	Fluoroscopy-guided POCS-guided	59 59 (identical cohort)	64.0% 54.0%	0.416

## Data Availability

All data for this study are stored in the Third Department of Internal Medicine, University of Toyama.

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
