# Peer review of "Peroral Cholangioscopy-Guided Targeted Biopsy versus Conventional Endoscopic Transpapillary Forceps Biopsy for Biliary Stricture with Suspected Bile Duct Cancer"

_jcm, 2022, doi:10.3390/jcm11020289_

Round 1
Reviewer 1 Report
jcm-1512033: Peroral cholangioscopy-guided targeted biopsy versus conventional endoscopic transpapillary forceps biopsy for biliary stricture with suspected bile duct cancer
This study compared the diagnostic abilities between peroral cholangioscopy (POCS)-guided biopsy sampling and fluoroscopy-guided biopsy for bile duct cancer. Because the diagnosis and management of biliary strictures still remain a challenge, this study is of great significance to present comparison data on the diagnostic method of biliary stricture. However, this study has several problems that need to be addressed .
<Major comments>
- It seems that the difference in the quantity and quality of tissue collection is closely related to the size of the biopsy forceps. Because the results may vary depending on the caliber of biopsy forceps, you should add in the discussion or modify conclusion that the results can be reversed with the development of the POCS-guided biopsy forceps.
- The POCS used in this study is SpyGlass Legacy or DS. Currently, the 3rd generation SpyGlass DS II has been commercialized, and new SpyBite Max biopsy forceps that can collect twice as much sample have been developed. If the new SpyBite Max can provide direct visualization and twice larger amount of tissue samples, it may possibly show better outcomes than ERCP-guided biopsy. Please clarify about this on discussion section including the difference of biopsy forceps between SpyBite Max and Radial Jaw 4P.
- In this article, sensitivity increased to 80% when POCS-guided biopsy and fluoroscopy-guided biopsy are used together. I wonder if your institution always performs ERCP guided biopsy after POCS-guided biopsy in biliary stricture. What is your institution’s diagnostic algorithm?
- Results may vary depending on the number of biopsy samplings. Please clarify the mean and standard deviation of the number of biopsy samples on both procedures.
- In previous study reported by Gerges et al. (Gastrointest Endosc 2020;91:1105-13), adverse events were comparable in POCS guided biopsy and ERCP guided brush sampling. In this study, POC and ERCP were performed simultaneously. Please clarify whether there were any adverse events during the procedures. If possible, compare the differences between the 2 procedures.
<Minor comments>
- Was there only one endoscopist who performed the procedure? If there were several endoscopists, it is necessary to describe the informations of endoscopists on Methods section.
- Supplementary description is needed whether the p-value of the pathological assessment (Table 4) was performed by the chi-square test, the Mann-whitney U test or other statistical method.
- It is unclear that the order of the procedure is disadvantageous for ERCP-guided biopsy. Because this study was not randomized, it seems to be a statistic limitation of this study. I recommend to add a proposal sentence for future randomized controlled trial in discussion section.
- Please clarify whether there was a difference in outcome according to the location of the proximal and distal bile ducts.
Author Response
December 29, 2021
Prof. Dr. Hidekazu Suzuki
Editor-in-Chief
Journal of Clinical Medicine
Gastroenterology & Hepatopancreatobiliary Medicine Section
Re: Manuscript ID: jcm-1512033
Title: Peroral cholangioscopy-guided targeted biopsy versus conventional endoscopic transpapillary forceps biopsy for biliary stricture with suspected bile duct cancer
Dear Prof. Dr. Hidekazu Suzuki
Thank you for your e-mail dated Dec 15, 2021, regarding the above manuscript. We greatly appreciate the opportunity to revise and resubmit this manuscript.
We have rewritten the manuscript taking into consideration the comments from the reviewers. We have attached the files of the revised manuscript and our point-by-point responses to the reviewers’ comments.
Thank you very much, once again, for allowing us to revise and resubmit the manuscript. We look forward to hearing from you regarding your evaluation of the revised manuscript.
With best regards,
Ichiro Yasuda, MD, PhD
Third Department of Internal Medicine,
University of Toyama,
2630 Sugitani, Toyama 930-0194, Toyama, Japan
Phone: +81 76 434 7301
Fax: +81 76 434 5027
E-mail: yasudaic@med.u-toyama.ac.jp
Responses to the comments by reviewer #1
Thank you for your valuable comments. Your comments helped us to improve the manuscript. Following your comments, we have revised the manuscript as below.
Your major comment #1:
It seems that the difference in the quantity and quality of tissue collection is closely related to the size of the biopsy forceps. Because the results may vary depending on the caliber of biopsy forceps, you should add in the discussion or modify conclusion that the results can be reversed with the development of the POCS-guided biopsy forceps.
Our response:
Thank you for suggesting good points. We totally agree with your comment. As you suggested, our results showed that the quality and quantity of tissue sample was closely correlated to the caliber (namely cup size) of biopsy forceps. Therefore, if the same size caliber forceps was available in both procedures, the sensitivity of POCS-guided biopsy would be better than that of conventional fluoroscopy-guided biopsy because more precise target biopsy is available in POCS-guided biopsy.
According to your recommendation, we revised the discussion and conclusions.
Please see the “Discussion” and “Conclusions” section (lines 252-261, 268-270, and 274-278).
Your major comment #2:
The POCS used in this study is SpyGlass Legacy or DS. Currently, the 3rd generation SpyGlass DS II has been commercialized, and new SpyBite Max biopsy forceps that can collect twice as much sample have been developed. If the new SpyBite Max can provide direct visualization and twice larger amount of tissue samples, it may possibly show better outcomes than ERCP-guided biopsy. Please clarify about this on discussion section including the difference of biopsy forceps between SpyBite Max and Radial Jaw 4P.
Our response:
As you suggested, novel SpyGlass DS system (SpyGlass DS II and SpyBite Max) have recently been commercialized and may improve the tissue sampling ability. We mentioned about the possibility in the discussion section.
Please see the “Discussion” section (lines 280-282).
Your major comment #3:
In this article, sensitivity increased to 80% when POCS-guided biopsy and fluoroscopy-guided biopsy are used together. I wonder if your institution always performs ERCP guided biopsy after POCS-guided biopsy in biliary stricture. What is your institution’s diagnostic algorithm?
Our response:
Our diagnostic algorithm of suspected bile duct cancer is as follow.
In cases suspected with biliary stricture by blood examinations and ultrasonography, multidetector-row computed tomography (MD-CT) was first performed. If the image findings were suspected with bile duct cancer, ERCP related procedures including pathological sampling and subsequent biliary drainage were attempted. If the MD-CT findings did not show any suspicious findings of unresectable and the distal end of the biliary stricture did not reach the papilla, POCS with targeted biopsy was basically attempted. At that time, fluoroscopy-guided biopsy was also performed following the POCS-guided biopsy, because our early experience of the POCS-guided targeted biopsy often showed failed diagnostic results and fluoroscopy-guided biopsy sometimes provided better results.
Please see the “Materials and Methods” section (lines 80-88), and also see the “Discussion” section (lines 243-248).
Your major comment #4:
Results may vary depending on the number of biopsy samplings. Please clarify the mean and standard deviation of the number of biopsy samples on both procedures.
Our response:
We agree with your comment. We also believe that the number of biopsy samples affect the diagnostic results. Our results showed that the number of samples were not different between the two biopsy procedures. The data had already shown in original Table 3. However, we presented the number of biopsy samples as median with range in original Table 3. Therefore, the values are replaced with mean with standard deviation in new Table 3 following your comment.
Your major comment #5:
In previous study reported by Gerges et al. (Gastrointest Endosc 2020;91:1105-13), adverse events were comparable in POCS guided biopsy and ERCP guided brush sampling. In this study, POC and ERCP were performed simultaneously. Please clarify whether there were any adverse events during the procedures. If possible, compare the differences between the 2 procedures.
Our response:
Cholangitis was observed in 5 cases (mild in 4 and moderate in 1) after the ERCP procedures. However, it was resolved in all cases by fasting and medication of antibiotics for 2 to 6 days. Post-ERCP pancreatitis did not occur in any cases. We added the data in the “Results”. However, we could not compare the adverse events between the two procedures, because the two procedures were performed in identical cases and during the same session.
Please see the “Results” section (lines 211-213).
Your minor comment #1:
Was there only one endoscopist who performed the procedure? If there were several endoscopists, it is necessary to describe the information of endoscopists on Methods section.
Our response:
All endoscopic procedures were performed by 4 experienced endoscopists with more than 1,000 ERCP experience (K.S., I.Y., S.D, and T.I). We additionally mentioned about it in the “Materials and Methods” section.
Please see the METHODs section (lines 91-92).
Your minor comment #2:
Supplementary description is needed whether the p-value of the pathological assessment (Table 4) was performed by the chi-square test, the Mann-Whitney U test or other statistical method.
Our response:
Our description about the statistical analysis was insufficient. The ordinal variables for pathological assessment were also tested using Mann-Whitney U test. We rewrote the description about the statistical analysis.
Please see “Statistical analysis” section (lines 159-164)
Your minor comment #3:
It is unclear that the order of the procedure is disadvantageous for ERCP-guided biopsy. Because this study was not randomized, it seems to be a statistic limitation of this study. I recommend to add a proposal sentence for future randomized controlled trial in discussion section.
Our response:
Following your advice, we added a proposal sentence for future randomized controlled trial in the “Discussion” section.
Please see “Discussions” section (lines 310-311).
Your minor comment #4:
Please clarify whether there was a difference in outcome according to the location of the proximal and distal bile ducts.
Our response:
Following your comment, we presented the diagnostic results according to the tumor location. The diagnostic sensitivity was not significantly different in distal and proximal bile duct in both of POCS- and fluoroscopy-guided biopsy (POCS-guided, distal 55.2% vs. proximal 52.4%; fluoroscopy-guided, distal 62.1% vs. proximal 66.7%).
Please see “Discussions” section (lines 290-293).
Thank you again for your valuable comments.

Reviewer 2 Report
Dear Editor, thank you so much for inviting me to revise this manuscript addressing a current topic.
The manuscript is quite well written and organized. English could be improved.
Figures and tables are comprehensive and clear. However, as you could see below, some points should be elucidated.
We suggest the following modifications:
- Introduction section: although the authors correctly included important papers in this setting, we believe a couple of studies should be cited within the introduction (PMID: 32606456 ; PMID: 33571059 ) only for a matter of consistency. We think it might be useful to introduce the topic of this study.
- In addition, we believe some issues deserve further discussion. In everyday clinical practice, we know that the pathologic confirmation of diagnosis is necessary before any non-surgical treatment and can be challenging in BTC, particularly in patients affected by primary sclerosing cholangitis and biliary strictures. In fact, decisions to undertake biopsies should follow a multidisciplinary discussion, especially in potentially resectable tumors. Moreover, endoscopic imaging and tissue sampling are useful but, sadly, biopsy samples are often inadequate for molecular profiling, and in addition, tissue sampling has reported high specificity but low sensitivity in diagnosis of malignant biliary strictures. Finally, the highly desmoplastic nature of BTC limits the accuracy of cytological and pathological approaches.
On the basis of these premises, in this scenario, it is urgent to develop new strategies in order to anticipate the diagnosis identifying BTC at an early, resectable stage, and to obtain sufficient material with which to perform genomic analysis. Among these strategies, liquid biopsy has received growing attention over the years, given the promising applications in cancer patients. More specifically, several studies have shown the potential role of liquid biopsy, and the authors should discuss this point, also reporting recent studies in this setting (doi: 10.3390/cells9030721; doi: 10.21873/cgp.20203).
- Methods and Statistical Analysis: nothing to add.
- Discussion section: Interesting section.
However, some changes and some additions are necessary.
Of note, the authors should expand the Discussion section, including a more personal perspective to reflect on. For example, they could answer the following questions – in order to facilitate the understanding of this complex topic to readers: what potential does this study hold? What are the knowledge gaps and how do researchers tackle them? How do you see this area unfolding in the next 5 years?
We think it would be extremely interesting for the readers, especially considering the challenging landscape of cholangiocarcinoma, where novel treatment options are opening the doors of a new world, with the hope to lower the recurrence rates of these aggressive malignancies.
One additional little flaw: the authors should better explain the limitations of their work, in the last part of the Discussion.
We believe some revisions are needed. The main strengths of this paper are that it addresses an interesting and very timely question and provides a clear answer, with some limitations.
We suggest a linguistic revision, the addition of some references for a matter of consistency, and some clarifications and extensive changes regarding some crucial points in everyday clinical practice of biliary tract cancers.
Author Response
December 29, 2021
Prof. Dr. Hidekazu Suzuki
Editor-in-Chief
Journal of Clinical Medicine
Gastroenterology & Hepatopancreatobiliary Medicine Section
Re: Manuscript ID: jcm-1512033
Title: Peroral cholangioscopy-guided targeted biopsy versus conventional endoscopic transpapillary forceps biopsy for biliary stricture with suspected bile duct cancer
Dear Prof. Dr. Hidekazu Suzuki
Thank you for your e-mail dated Dec 15, 2021, regarding the above manuscript. We greatly appreciate the opportunity to revise and resubmit this manuscript.
We have rewritten the manuscript taking into consideration the comments from the reviewers. We have attached the files of the revised manuscript and our point-by-point responses to the reviewers’ comments.
Thank you very much, once again, for allowing us to revise and resubmit the manuscript. We look forward to hearing from you regarding your evaluation of the revised manuscript.
With best regards,
Ichiro Yasuda, MD, PhD
Third Department of Internal Medicine,
University of Toyama,
2630 Sugitani, Toyama 930-0194, Toyama, Japan
Phone: +81 76 434 7301
Fax: +81 76 434 5027
E-mail: yasudaic@med.u-toyama.ac.jp
Responses to the comments by reviewer #2
Thank you for your favorable and helpful comments for improving our manuscript. Following your comments, we have revised our manuscript as follows:
Your comment #1:
Introduction section: although the authors correctly included important papers in this setting, we believe a couple of studies should be cited within the introduction (PMID: 32606456 ; PMID: 33571059 ) only for a matter of consistency. We think it might be useful to introduce the topic of this study.
In addition, we believe some issues deserve further discussion. In everyday clinical practice, we know that the pathologic confirmation of diagnosis is necessary before any non-surgical treatment and can be challenging in BTC, particularly in patients affected by primary sclerosing cholangitis and biliary strictures. In fact, decisions to undertake biopsies should follow a multidisciplinary discussion, especially in potentially resectable tumors. Moreover, endoscopic imaging and tissue sampling are useful but, sadly, biopsy samples are often inadequate for molecular profiling, and in addition, tissue sampling has reported high specificity but low sensitivity in diagnosis of malignant biliary strictures. Finally, the highly desmoplastic nature of BTC limits the accuracy of cytological and pathological approaches.
Our response:
Thank you for introducing two valuable papers. After reading the papers, we rewrote the “Introduction” section and added the papers as references.
Please see the “Introduction” section (lines 41-48) and References (#1-2).
Your comment #2:
Of note, the authors should expand the Discussion section, including a more personal perspective to reflect on. For example, they could answer the following questions – in order to facilitate the understanding of this complex topic to readers: what potential does this study hold? What are the knowledge gaps and how do researchers tackle them? How do you see this area unfolding in the next 5 years?
Our response:
Following your comments, we briefly mentioned about the potential of this study, our proposal about the biopsy techniques and the necessity of RCT, and near future perspective in this field.
Please see the “Discussion” section (lines 216-311).
Your comment #3:
One additional little flaw: the authors should better explain the limitations of their work, in the last part of the Discussion.
Our response:
Following your comments, we additionally mentioned about the limitations of this study.
Please see “Discussions section” (lines 294-311)
Your comment #4:
We believe some revisions are needed. The main strengths of this paper are that it addresses an interesting and very timely question and provides a clear answer, with some limitations. We suggest a linguistic revision, the addition of some references for a matter of consistency, and some clarifications and extensive changes regarding some crucial points in everyday clinical practice of biliary tract cancers.
Our response:
Thank you for your favorable comments. We rewrote the text through whole the manuscript and also took linguistic revision by native English speaker.
Thank you again for your valuable comments.

Round 2
Reviewer 2 Report
The authors modified the manuscript according to our suggestions.
We recommend Acceptance.